# mRNA vaccines encoding influenza virus hemagglutinin (HA) elicits immunity in mice from influenza A virus challenge

Z. Beau Reneer[1]*, Harrison C. Bergeron[1], Stephen Reynolds[1], Elena Thornhill-Wadolowski[2], Lan Feng[2], Marcin Bugno[2], Agnieszka D. Truax[2], Ralph A. Tripp[1]

1 Department of Infectious Diseases, College of Veterinary Medicine, University of Georgia, Athens, GA, Unites States of America, 2 Immorna Biotherapeutics, Morrisville, NC, United States of America

* zbreneer@unc.edu

**Data Availability Statement:** I have uploaded the data to 'Open Science Framework' DOI 10.17605/OSF.IO/S6Z7X.

## Abstract

Influenza viruses cause epidemics and can cause pandemics with substantial morbidity with some mortality every year. Seasonal influenza vaccines have incomplete effectiveness and elicit a narrow antibody response that often does not protect against mutations occurring in influenza viruses. Thus, various vaccine approaches have been investigated to improve safety and efficacy. Here, we evaluate an mRNA influenza vaccine encoding hemagglutinin (HA) proteins in a BALB/c mouse model. The results show that mRNA vaccination elicits neutralizing and serum antibodies to each influenza virus strain contained in the current quadrivalent vaccine that is designed to protect against four different influenza viruses including two influenza A viruses (IAV) and two influenza B (IBV), as well as several antigenically distinct influenza virus strains in both hemagglutination inhibition assay (HAI) and virus neutralization assays. The quadrivalent mRNA vaccines had antibody titers comparable to the antibodies elicited by the monovalent vaccines to each tested virus regardless of dosage following an mRNA booster vaccine. Mice vaccinated with mRNA encoding an H1 HA had decreased weight loss and decreased lung viral titers compared to mice not vaccinated with an mRNA encoding an H1 HA. Overall, this study demonstrates the efficacy of mRNA-based seasonal influenza vaccines are their potential to replace both the currently available split-inactivated, and live-attenuated seasonal influenza vaccines.

## Introduction

There are four types of influenza viruses, i.e. A, B, C, and D. Influenza A virus (IAV) and influenza B virus (IBV) can cause seasonal epidemics of disease in people every winter in the United States [1]. IAV is the only influenza virus type known to cause pandemics. IAVs are divided into subtypes based on HA and neuraminidase (NA) gene expression. There are 18 HA subtypes and 11 NA subtypes where IAV subtypes exist mostly in wild birds [2]. The IAV subtypes that routinely circulate in people include A(H1N1) and A(H3N2). Humans are regularly infected with IAVs and IBVs which cause approximately 250,000–650,000 deaths annually

**Funding:** Immorna provided support in the form of salaries for author Z. Beau Reneer but did not have any additional role in the study design, data collection and analysis, decision to publish, or preparation of the manuscript. The specific roles of these authors are articulated in the 'author contributions' section.

**Competing interests:** Our relationship with Immorna Biotherapeutics does not alter our adherence to PLOS ONE policies on sharing data and materials.

despite widely available seasonal influenza vaccines [3]. Seasonal influenza vaccines contain four components, i.e. two IAV strains (H1N1 subtype and H3N2 subtype) and two IBV strains (B/Victoria lineage and B/Yamagata lineage) [4]. The effectiveness of seasonal influenza vaccines varies each season but is typically only 40–60% effective. However, the effectiveness of the seasonal influenza vaccines can be lower than 20% [5].

Given the success of the SARS-CoV-2 mRNA vaccine for reducing COVID-19 [6], recent studies have evaluated mRNA-lipid nanoparticle (LNP) influenza virus vaccines. These studies have evaluated mRNA vaccines containing multiple IAV or IBV HA antigens [7–9]. These studies have evaluated multiple different doses of mRNA-based vaccines and various formulations of nucleoside modified mRNA which all elicited antibody titers to their encoded antigens. In this study, we evaluated a novel quadrivalent mRNA vaccine, produced my Immorna Biotherapeutics, for seasonal influenza viruses in BALB/c mice.

The results in our study show that mRNA vaccination elicits potent neutralizing serum antibodies to all IAV and IBV strains that are currently in the three influenza vaccines that are recommended for people 65 years and older, i.e. Fluzone High-Dose Quadrivalent® vaccine, Flublok Quadrivalent® recombinant influenza vaccine, and Fluad Quadrivalent® adjuvanted flu vaccine. The WHO recommends quadrivalent vaccines for use in the 2022–2023 season for egg-based vaccines consisting of an A/Victoria/2570/2019 (H1N1) pdm09-like virus, an A/Darwin/9/2021 (H3N2)-like virus, a B/Austria/1359417/2021 (B/Victoria lineage-like), and a B/Phuket/3073/2013 (B/Yamagata lineage-like) viruses. For cell culture or recombinant vaccines, the WHO recommends a vaccine consisting of A/Wisconsin/588/2019 (H1N1 pdm09-like), A/Darwin/6/2021 (H3N2-like virus, B/Austria/1359417/2021 (B/Victoria lineage-like), and a B/Phuket/3073/2013 (B/Yamagata lineage-like) viruses. Low (5 µg) and high-dose (20 µg) quadrivalent mRNA vaccines produced antibody titers in mice comparable to the antibodies elicited by the monovalent mRNA vaccines in both hemagglutinin inhibition (HAI) and neutralization assays. Mice vaccinated with mRNA encoding an H1 HA (e.g. A/Wisconsin/588/2019) alone or a quadrivalent mRNA vaccine containing an H1 HA component had decreased weight loss and decreased lung viral titers compared to mice vaccinated with an H3 HA mRNA vaccine. The findings show that a multivalent mRNA influenza vaccine protects against matched and antigenically distinct mismatched viruses by inducing neutralizing antibodies.

## Results

### mRNA vaccination and challenge with A/California/04/2009 (A/Cal/09) H1N1

Female BALB/c mice (n = 6) were vaccinated with either an mRNA vaccine coding for a single HA, or the quadrivalent vaccine including 4 mRNAs combined at 1:1:1:1 weight ratio, each coding for 1 of 4 different HAs from 2022–2023 seasonal influenza viruses. Specifically, a quadrivalent vaccine consisting of A/Wisconsin/588/2019 (A/Wis/19 H1N1), A/Darwin/6/2021 (A/Dar/21 H3N2), B/Austria/1359417/2021 (B/Aus/21 Victoria lineage), and B/Phuket/3073/2013 (B/Phu/14 Yamagata lineage). The vaccine groups are outlined in Table 1 where Group 1 = lipid nanoparticle (LNP) only, Group 2 = A/Wis/19, Group 3 = A/Dar/21, Group 4 = B/Aus/21, Group 5 = B/Phu/13, Group 6 = quadrivalent (20 ug), Group 7 = quadrivalent (5 ug), Group 8 = quadrivalent (20 ug, single vaccination), Group 9 = quadrivalent (5 ug, two site injections—one site for IAV and one site for IBV). mRNA vaccination was examined two months post boost vaccination, and the mice were challenged with a sub-lethal dose of H1N1 A/California/04/2009 (A/Cal/09). At days 2 and 3 post-infection, vaccine Groups 1 (LNP only), 3 (A/Dar/21, H3N2), group 4 (B/Aus/21) and 5 (B/Phu/13) had significantly (p<0.05)

**Table 1. Vaccination outline.**

| Group Number | Description | Dosage |
|---|---|---|
| Group 1 | LNP only | 2 injections |
| Group 2 | A/Wisconsin/588/2019 | 5 µg/dose (2 injections) |
| Group 3 | A/Darwin/6/2021 | 5 µg/dose (2 injections) |
| Group 4 | B/Austria/1359417/2021 | 5 µg/dose (2 injections) |
| Group 5 | B/Phuket/3073/2013 | 5 µg/dose (2 injections) |
| Group 6 | Quadrivalent | 20 µg/dose (2 injections) |
| Group 7 | Quadrivalent | 5 µg/dose (2 injections) |
| Group 8 | Quadrivalent | 20 µg/dose (1 injection) |
| Group 9 | Quadrivalent | 5 µg/dose (2 injections IAVs in right thigh and IBVs in left thigh) |

lower body weights than any of the other vaccine groups (Fig 1). At day 5 post-infection, the remaining mice in each group were euthanized and their spleens were harvested for cellular analysis.

## Lung virus titers

Mice (n = 3/Group) were sacrificed on day 3 post-infection and their lungs were harvested to quantify viral titers (Fig 2). Groups 4 (B/Aus/21) and 5 (B/Phu/13) were not included in lung viral titer analysis due to insufficient numbers of mice in each group. A mouse in each group had to be sacrificed prior to infection due to pre-infection bleeds. Vaccine Groups 1 (LNP only) and 3 (A/Dar/21) both had greater than $10^6$ PFU/mL lung viral titers. Groups 7 (quadrivalent, 5 ug), 8 (quadrivalent, 20 ug) and 9 (quadrivalent, 5 ug, 2 injections—IAVs in right thigh and IBVs in left thigh) all had approximately $10^4$ PFU/mL lung viral titers, while Groups 2 (A/Wis/19) and 6 (quadrivalent, 20 ug, 2 injections) had viral lung titers below the limit of detection of the plaque assay (50 PFU/mL). Both Groups 1 (LNP only) and 3 (A/Dar/21) had significantly (p<0.05) higher lung viral titers compared to Group 2 (A/Wis/19), Group 6 (quadrivalent, 20 ug), and Group 7 (quadrivalent, 5 ug) while Groups 8 (quadrivalent, 20 ug) and 9 (quadrivalent, 5 ug, 2 injections IAVs in right thigh and IBVs in left thigh) had significantly (p<0.05) higher lung viral titers compared to Groups 2 and 6.

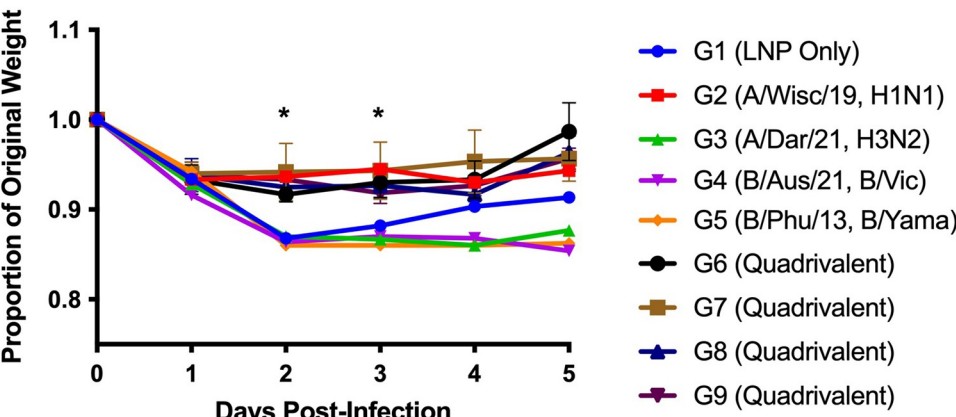

**Fig 1. A/California/04/2009 weight loss.** Weight loss of A/California/04/2009 challenged BALB/c mice (n = 6 mice/group). BALB/c mice were challenged intranasally with $10^4$ PFU/mouse. Weight loss was recorded for five days post-infection. Asterisks (*) represent statistical significance based on two-way ANOVA (p<0.05).

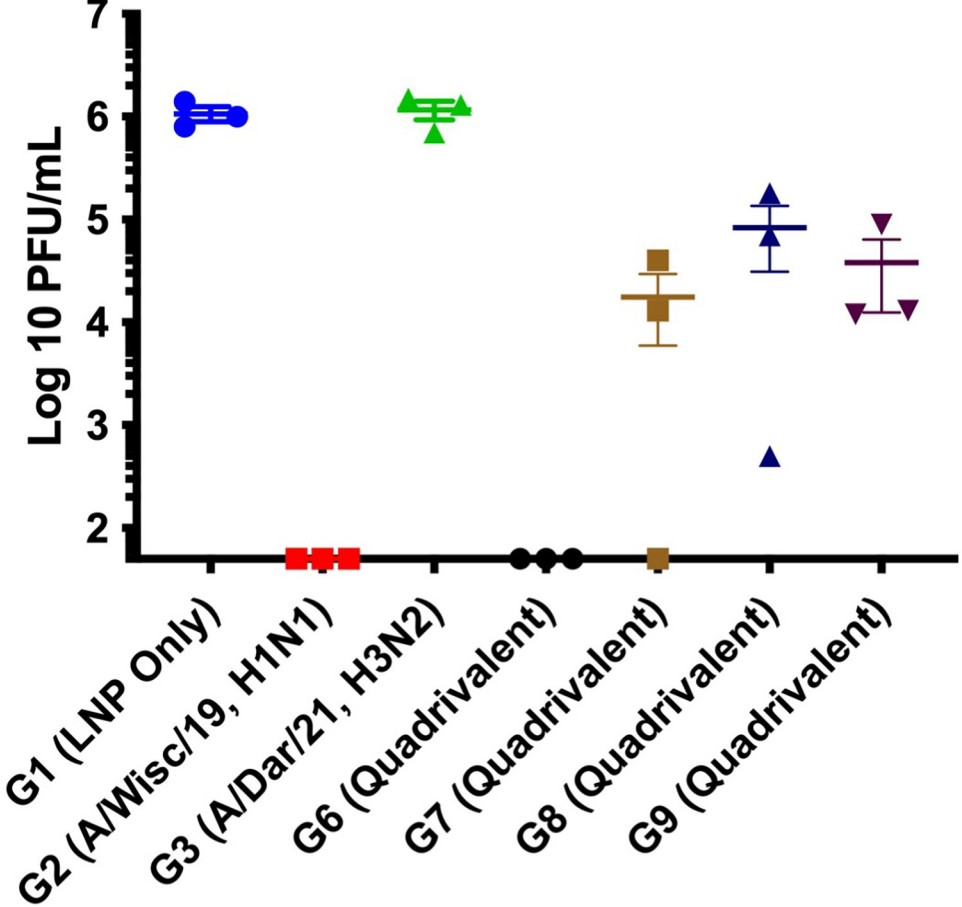

**Fig 2. Lung viral titers.** Viral titers in lungs of vaccinated mice. Mice were infected with A/California/04/2009 i.n. with $10^4$ PFU/mouse. Mice from each group (n = 3) were sacrificed on day 3 pi and lungs were harvested using plaque assays as described. Statistical significance was determined by one-way ANOVA (p<0.05).

## Hemagglutination inhibition (HAI) antibodies

Serum samples were collected on day 19 and 42 post-prime vaccination (or day 21 post-boost). The sera at each time point was titrated for receptor-blocking antibodies by HAI assay against a panel of four viruses matching the HA sequence encoded by the various mRNAs used in the vaccines (Fig 3). The day 42 sera (day 21 post-boost) were also tested against nine influenza viruses having antigenically distinct HA sequences for each influenza subtype. On day 19, the A/Vic/19, A/Dar/21, and A/Aus/19 single mRNA vaccinated mice had substantially high average HAI titers (>1:40) to each of their homologous viruses (Fig 3A–3C). Each of the four quadrivalent mRNA vaccinated groups had high averaged HAI titers (>1:40) to A/Vic/19 (Fig 3A). High-dose (20 μg/dose) quadrivalent vaccine groups, (i.e., Groups 6 and 8) also had high averaged HAI titers (>1:40) to both A/Dar/21 and A/Aus/19 (Fig 3B and 3C). None of the vaccine groups had high average HAI titers (>1:40) to B/Phu/13) (Fig 3D). For A/Vic/19, Groups

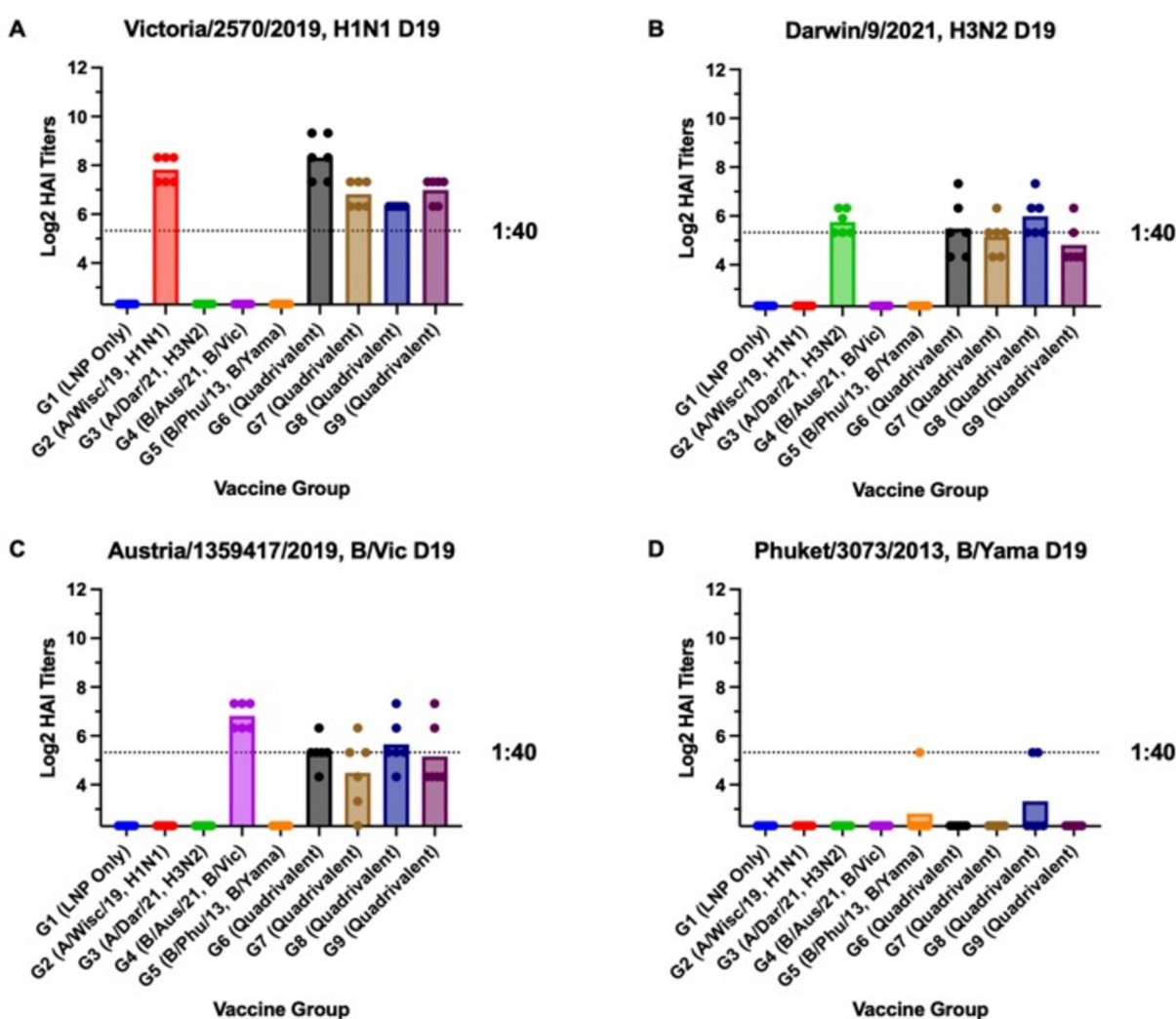

**Fig 3. Day 19 HAI titers.** Cross-reactive antibody responses on day 19 post-vaccination. HAI titers are shown for each vaccine group against the four influenza viruses used in the vaccine. Serum from each mouse was obtained on day 19 post-vaccination. Dotted lines indicate 1:40 HAI titers. Statistical significance was determined by one-way ANOVA (p<0.05).

2 and 6 had significantly (p<0.05) higher HAI titers than any of the other vaccine groups (Fig 3A). For A/Dar/21, Groups 3, 6, 7, 8, and 9 all had significantly higher (p<0.05) HAI titers than any of the other vaccine groups (Fig 3B). Group 4 (B/Aus/19) had significantly (p<0.05) higher average HAI titers than all of the vaccine groups with the exception of Group 8 (quadrivalent 20 ug/dose). There were no significant differences between any of the vaccination groups for the B/Phu/13 virus (Fig 3D).

The day 42 sera (day 21 post-boost) from Groups 2, 6, 7, and 9 all had high average HAI titers (>1:40) to each of the H1N1 viruses (A/Vic/19, A/GM/19, A/ Bris/18), and A/Cal/09 (Fig 4A, 4E, 4I and 4L). Group 8 had high average HAI titers (>1:40) to the A/Vic/19 and A/Bris/18 (Fig 4A and 4I), and Group 3 had high average HAI titers (>1:40) to each of the H3N2 viruses (A/Dar/21), A/Tas/20, A/HK/19, and A/Kan/17 (Fig 4B, 4F, 4J and 4M). Groups 6, 7, 8, and 9 all had high average HAI titers (>1:40) to A/Dar/21 and A/Tas/20. Group 6 also had high average HAI titers (>1:40) to A/Kan/17. Groups 4, 6, 7, 8, and 9 had high average HAI titers (>1:40) to B/Aus/19 (Fig 4C). None of the vaccine groups induced an average HAI titer

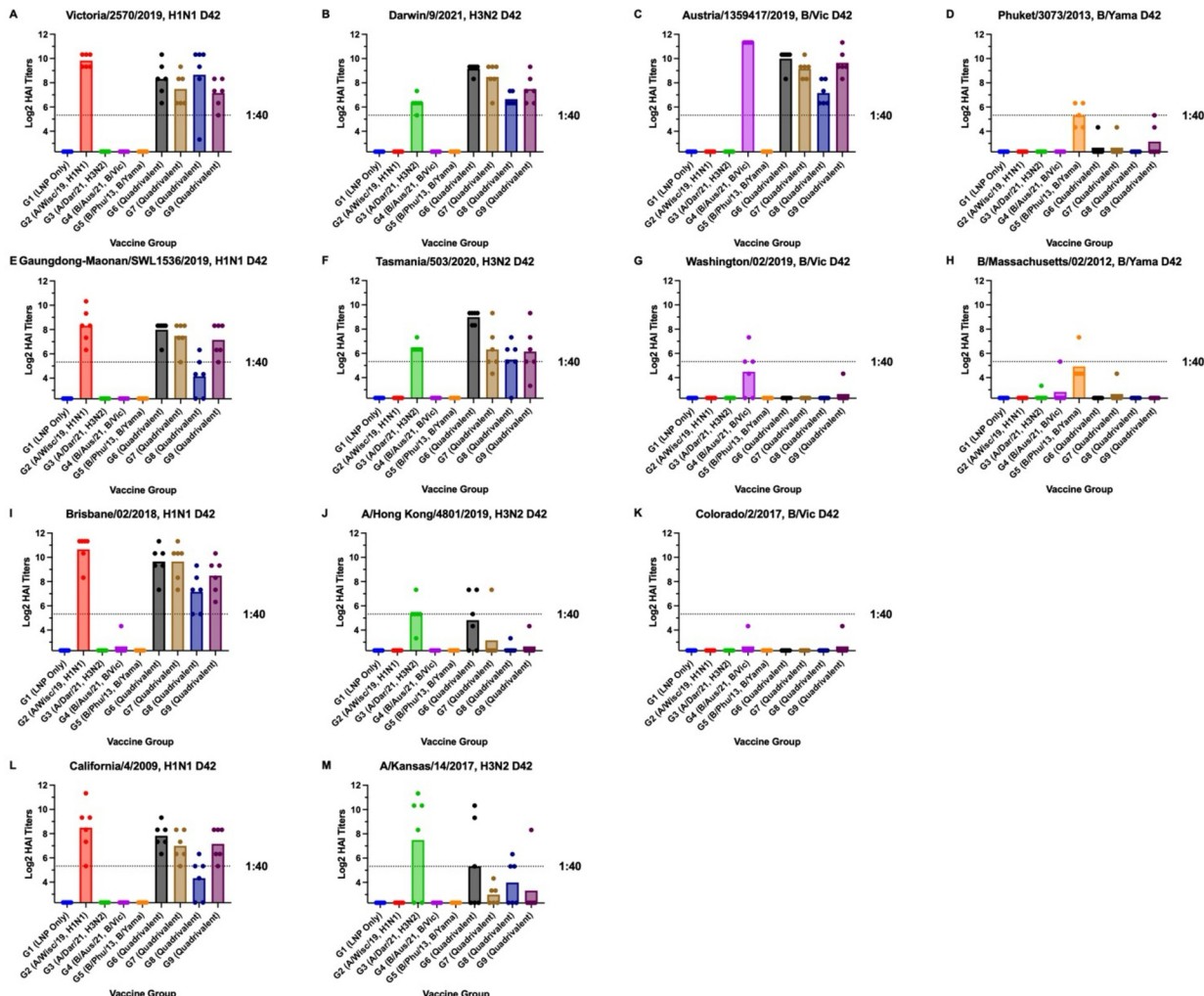

**Fig 4. Day 42 HAI titers.** Cross-reactive antibody responses on day 42 post-vaccination. HAI titers are shown for each vaccine group against the four influenza viruses used in the vaccine as well as antigenic variants for each influenza virus subtype. Panels a, e, i and l are H1N1 viruses. Panels b, f, j and m are H3N2 viruses. Panels c, g and k are B/Victoria influenza viruses while panels d and h are B/Yamagata viruses. Serum from each mouse was obtained on day 42 post-vaccination. Dotted lines indicate 1:40 HAI titers. Error bars represent standard mean errors. Statistical significance was determined by one-way ANOVA ($p < 0.05$).

of 1:40 to either B/Wash/19 or B/Col/17 (Fig 4G and 4K). Group 5 had a high average HAI titer (>1:40) to B/Phu/13 and B/Mass/12 (Fig 4D and 4H).

On day 42, Group 2 sera had significantly ($p < 0.05$) higher average HAI titers than Groups 1, 3, 4, 5, 7, and 9 for A/Vic/19 (Fig 4A). Group 2 had significantly ($p < 0.05$) higher average HAI titers than Groups 1, 3, 4, 5, and 8 for A/GM/19 and A/Cal/09 (Fig 4E and 4L), while Group 2 had significantly ($p < 0.05$) higher average HAI titers than Groups 1, 3, 4, 5, 8 and 9 for A/Bris/18 (Fig 4I). For A/Dar/21, Group 3 had significantly ($p < 0.05$) lower HAI titers than Groups 6, 7, and 9 (Fig 4B). Group 3 had significantly ($p < 0.05$) higher HAI titers than Groups 1, 2, 4, 5, and 6 for the A/Tas/20 virus (Fig 4F) while Group 3 had significantly ($p < 0.05$) higher average HAI titers than Groups 1, 2, 4, 5, 7, 8 and 9 for A/HK/19 virus (Fig 4J). For A/Kan/17, Group 3 had significantly higher ($p < 0.05$) HAI titers than Groups 1, 2, 4, 5, 7, and 9 (Fig 4M). For both the A/Aus/21 and B/Wash/19, Group 4 had significantly ($p < 0.05$) higher average HAI titers than any of the other vaccine groups while for the B/Phu/13 and B/Mass/12 viruses

**Table 2. Mouse neutralization titers.**

|  | A/Vic/19 | A/Dar/21 | B/Aus/19 | B/Phu/13 | A/Cal/09 |
|---|---|---|---|---|---|
| Group 1 | 5 | 5 | 5 | 5 | 5 |
| Group 2 | 560 | 7 | 5 | 6 | 640 |
| Group 3 | 40 | 594 | 5 | 6 | 8 |
| Group 4 | 8 | 5 | 640 | 24 | 5 |
| Group 5 | 7 | 6 | 25 | 555 | 5 |
| Group 6 | 622 | 507 | 640 | 453 | 640 |
| Group 7 | 640 | 293 | 640 | 199 | 640 |
| Group 8 | 640 | 50 | 640 | 95 | 480 |
| Group 9 | 640 | 196 | 640 | 380 | 525 |

(Fig 4D and 4H), Group 5 had significantly ($p<0.05$) higher average HAI titers than any of the other vaccine groups. Overall, the HAI titers against the A/Cal/09 correlated with protection from disease.

## Neutralization assays

The day 42 post-prime vaccination sera were tested using a neutralization assay (Table 2). Group 2 sera from A/Wis/19 had significantly ($p<0.05$) higher neutralization titers than Groups 1, 3, 4, and 5 against the A/Vic/19 virus, while Group 2 had significantly ($p<0.05$) higher neutralization titers than Groups 1, 3, 4, 5, and 8 against A/Cal/09. Group 3 sera from A/Dar/21 had significantly ($p<0.05$) higher neutralization titers than sera from Groups 1, 2, 4, 5, 7, 8, and 9. Group 4 sera from B/Aus/21 had significantly ($p<0.05$) higher neutralization titers than sera from Groups 1, 3, 4, and 5 while sera from Group 5 (B/Phu/13) had significantly ($p<0.05$) higher neutralization titers than every group tested except for sera from Group 6 (quadrivalent, 20 ug). Overall, the neutralization titers against the A/Cal/09 virus correlated with protection from disease.

## Discussion

The development of a universal influenza vaccine that confers broad and durable protection against a diverse range of circulating influenza viruses is a pursued goal. HA is a major protective antigen recognized by Ab in the response to influenza virus infection or vaccination. There are 18 known IAV HA subtypes and two IBV HA subtypes making a universal vaccine technically and immunologically challenging with the current live and inactivated vaccine technology [10]. Conventional seasonal inactivated influenza vaccines include three (trivalent) or four (quadrivalent) HA subtypes and elicit largely strain-specific immune responses against the HA subtypes included in the vaccine. During the last two decades, there has been broad interest in RNA-based technologies for the development of prophylactic and therapeutic vaccines and several preclinical and clinical studies have shown that mRNA vaccines can provide a safe and effective immune response in animal models and humans [10, 11]. Influenza mRNA vaccines are composed of a synthetic mRNA template encoding a dominant viral antigen, e.g. the HA. The HA mRNA is translated to protein following delivered into a cell, however, to achieve efficient delivery the mRNA is modified to improve its stability, prevent degradation, and enhance translation, and this is encapsulated within a lipid nanoparticle (LNP) [12]. A key advantage of mRNA vaccines compared to conventional vaccines is that they allow for faster and more scalable vaccine production.

In this study, we examined an influenza mRNA vaccine by incorporating the mRNAs from four different influenza viruses including two circulating IAV and two IBVs, where mice were

intramuscularly (i.m.) immunized and/or boosting mice, and the serum Ab responses to similar and antigenically distinct influenza virus strains in both HAI and neutralization assays determined. The results showed that mRNA vaccination elicited binding and potent neutralizing serum Abs at day 42 post-vaccination to homologous (matched) HA to all IAV and IBV strains tested. We also evaluated a quadrivalent mRNA vaccine such as one recommended for use in the 2022–2023 season in egg-based vaccines, i.e. A/Wis/19 (H1N1), A/Dar/21, (H3N2), B/Aus/19 (B/Victoria-like), and B/Phu/13 (B/Yamagata-like). The quadrivalent mRNA vaccine we examined consisted of mRNA coding for the HA of A/Vic/19 (H1N1) A/Dar/21 (H3N2), B/Aus/21 (B/Victoria lineage), and B/Phu/13 (B/Yamagata lineage). The rationale was that quadrivalent vaccines are recommended for people 65 years and older, i.e. Fluzone High-Dose Quadrivalent® vaccine, Flublok Quadrivalent® recombinant influenza vaccine, or Fluad Quadrivalent® adjuvanted flu vaccine [13]. The quadrivalent mRNA vaccines generated antibody titers comparable to the antibodies elicited by the monovalent vaccines to each tested virus. These results showed that the mRNA influenza vaccines induce strain-specific anti-HA responses against each subtype rather than eliciting antibodies that are broadly cross-reactive against a range of HAs. The HAI responses to B/Phuket/3073/2013 were lower than the other antigens. Upon reviewing the mRNA sequence, we determine that there is a single amino acid mutation in a known antigenic site that is likely causing this depressed response.

mRNA requires an effective delivery platform to protect the nucleic acid from degradation and allow for cellular uptake. Lipid nanoparticles (LNP) have been to successfully deliver mRNA, particularly as an LNP-mRNA vaccine in the fight against COVID-19 [14]. For example, two successful COVID-19 vaccines mRNA-1273 [15] and BNT162b [16] use LNPs to deliver mRNA as a SARS-CoV-2 vaccine. Likewise, mRNA-1440 and mRNA-1851 vaccines are LNP-mRNA encoding HA from H10N8 and H7N9 influenza viruses, respectively [16, 17]. The vaccine efficacy was evaluated by HAI and neutralization assays where it was shown that a 100 µg dose induced 78% and 87.0% HAI and neutralization, respectively for H10N8, and a 50 µg dose resulted in 96 and 100% neutralization seroconversion for H7N9 [18]. LNP-mRNA vaccines expressing influenza HA have been shown to enhance Ab and T cell activation [9, 19]. mRNA encapsulated in LNP induces a potent cellular and humoral immune response and facilitates the delivery of mRNA for vaccines against several viruses. The features of LNP-mRNA vaccine formulations have been preclinically and clinically investigated which has allowed for the rapid development and clinical use of the COVID-19 mRNA vaccines. Translating the SARS-CoV-2 LNP-mRNA vaccine platform to other viruses is under investigation however instability and the short half-life of mRNA along with potential safety and storage concerns need to be determined before clinical use.

We explored different situations in our mRNA vaccine delivery to mice such as dose, e.g. 5 or 20 µg, boosting, and vaccine injection site (e.g. left or right thigh). The quadrivalent mRNA vaccines generated Ab titers comparable to the monovalent vaccines to each tested virus in both HAI and neutralization assays. The average HAI titers for each vaccine group far exceeded 1:40 to each homologous virus which has been correlated with a 50% reduction in influenza like illness (ILI) in humans and is widely considered to be the minimum protective titer for influenza vaccines [20].

In this study, mice were vaccinated with a mRNA vaccine encoding an H1 HA alone or with quadrivalent mRNA vaccines each containing an H1 HA component. Vaccinated mice had decreased weight loss (disease) and decreased lung viral titers compared to mice not vaccinated with an mRNA encoding an H1 HA. Largely, 5 or 20 ug mRNA vaccination resulted in similarly robust Ab titers when mice were boosted with homologous mRNA vaccines. A single vaccination with 20 µg of quadrivalent mRNA vaccination was mostly equivalent to HAI and neutralization Ab titers of HA derived from IAVs, i.e. A/Vic/19 and A/Dar/21, as well as A/

Cal/09, A/GM/19, A/Bris/18, A/Tas/20, A/HK/19, and A/Kan/17. These findings show that 5 or 20 µg of mRNA vaccine encoding H1 HA can induce robust Ab responses to H1N1 and H3N2 IAVs, IBVs, and induce similar levels of B and T cells in the spleen of primed mice at day 5 post-challenge with A/California/04/2009.

## Materials and methods

### Viruses growth

A/California/04/2009 (H1N1 A/Cal/09), A/Victoria/2570/2019 (H1N1 A/Vic/19), A/Guang-dong-Maonan/SWL1536/2019 (H1N1 A/GM/19), A/Brisbane/02/2018 (H1N1 A/Bris/18), A/Darwin/9/2021 (H3N2 A/Dar/21), A/Tasmania/503/2020 (H3N2 A/Tas/20), A/Hong Kong/4801/2019 (H3N2 A/HK/19), A/Kansas/14/2017 (H3N2 A/Kan/17), B/Austria/1359417/2019 (B/Aus/19 Victoria lineage), B/Washington/02/2019 (B/Was/19 Victoria Lineage), B/Colorado/6/2017 (B/Col/17 Victoria lineage), B/Phuket/3073/2013 (Yamagata lineage) and B/Massachusetts/02/2012 (B/Phu/13 Yamagata lineage) were obtained from either BEI Resources (Manassas, Virginia) or provided by Dr. Ted Ross at the University of Georgia. Each virus was passaged using Madin-Darby kidney (MDCK) cells as described [21]. Each virus was harvested and aliquoted into tubes that were stored at -80˚C. Each virus was titrated using a standard influenza plaque assay explained below.

**Mouse vaccinations and challenge.** Female BALB/c mice (6–8 weeks old) were purchased from Jackson Laboratory (Bar Harbor, ME). The mice were housed in microisolator units and fed *ad libitum*. The mice were vaccinated with either 5 µg or 20 µg of mRNA per dose with a lipid-nanoparticle (LNP) dispersion provided by Immorna Biotherapeutics (Morrisville, North Carolina) in a total volume of 100 µL (50 µL/hind quarter). The mice were boosted with the same vaccine formulation and with the same dosage at three weeks post-prime vaccination except for the Group 8 vaccine group which only received one vaccination on day 0. The 'mock' vaccinated mice were administered LNP without mRNA. Six mice were included in each vaccination group for a total of 54 mice.

Blood was obtained from the mice by cheek bleeds and collected in blood collection tubes (Sarstedt, Newton, North Carolina) on days 19 and 42 post-vaccination. The blood was centrifuged at 500 xG for seven minutes at room temperature (RT). Serum samples were transferred to new 1.5 mL microcentrifuge tubes and stored at -20˚C. Mice were challenged intranasally (i.n.) 2 months post final vaccination. The mice were intraperitoneally (i.p.) anesthetized using Avertin (2,2,2-tribromoethanol) (Sigma Aldrich, St. Louis, MO) and i.n. infected with $10^4$ PFU/mouse of A/California/04/2009 diluted in PBS (Gibco). A clinical sign scoring system was used that included lethargy (n = 1), dyspnea (n = 2), body weight loss <15–20% of original weight, and body weight loss ≥20% of original body weight (n = 3). Any mouse that achieved a score of 3 or higher would be euthanized. No mouse achieved a clinical score requiring euthanasia during this study. Three mice from each group were humanely sacrificed by overdose of euthanasia and cervical dislocation on day 3 post-infection, and their lungs were harvested. The lungs were transferred to gentleMACS tubes (Miltenyi Biotech, Westphalia, Germany) containing 1 mL of DMEM (Gibco) supplemented with antibiotic/antimycotic (Gibco). The tubes were kept on ice until the lungs were homogenized. After homogenization, the lungs were centrifuged (500 xG, 3 min) to pellet the cellular debris, and the supernatants were transferred into fresh 1.5 mL microcentrifuge tubes. All lung homogenates were stored at -80˚C to prevent degradation of the virus.

All challenged mice were weighed daily. None of the mice exceeded 20% weight loss from its original weight. No severe clinical symptoms (with a clinical score 3 or higher) requiring the animals to be humanely euthanized were observed. All procedures were performed in

accordance with the University of Georgia institutional animal care and use committee (IACUC approval 03-006-Y2-A0).

**Hemagglutination inhibition (HAI) assay.** The hemagglutination inhibition (HAI) assay was used to quantify antibodies that bind at or near the receptor-binding site on the HA protein by measuring the inhibition in the agglutination of turkey red blood cells (RBCs) as described [22] and was adapted from the WHO laboratory of influenza surveillance manual. To inactivate nonspecific inhibitors, the sera were treated with a receptor-destroying enzyme (RDE) (Denka Seiken Company, Chuo, Tokyo) prior to being tested. Briefly, three parts RDE were added to one-part sera and incubated overnight at 37°C. RDE was inactivated by incubating the serum-RDE mixture at 56°C for approximately 45 min and following incubation six parts PBS was added to the RDE-treated sera. RDE-treated sera were two-fold serially diluted in U-bottom microtiter plates (Corning, New York). An equal volume of each virus was adjusted to approximately 8 hemagglutination units (HAU) per 0.025 mL and was added to each well of the U-bottom microtiter plates (Corning). The plates were covered and incubated at RT for 20 min before adding 0.05 mL of RBCs which were allowed to settle for 30 min at RT.

The HAI titer was determined from the reciprocal dilution of the last well that contained non-agglutinated RBCs. Positive and negative serum controls were included on each plate. At the beginning of the study, mice were negative (HAI titer of <1:10) for antibodies to each of the four viruses used in the vaccines. Seroprotection was defined as a HAI titer ≥1:40.

**Determination of viral lung titers.** Lung homogenates were thawed then 10-fold serial dilutions of the lung homogenates were overlaid on MDCK cells. The MDCK cells were at >90% confluency at the time that the assay was performed. Lung homogenate samples were incubated for 60 min at RT with gentle mixing every 15 min. After 60 min, the serial dilutions were removed and the MDCK cells were washed with serum-free Dulbecco's Modified Eagle Medium (DMEM, Gibco, Thermo Fisher). The medium was removed and replaced with 1 mL of a mixture of plaque media and 2.0% agarose (Sigma, St. Louis, MO). Plaque media contained MEM, HEPES, and L-Glutamine (all from Gibco). The MDCK cells were incubated at 37°C with 5% $CO_2$ for 48h. After 72h, the agarose overlay was removed and the cells were washed with PBS. MDCK cells were fixed with 10% neutral buffered formalin (Fisher Scientific) for 15 min. The formalin was subsequently discarded, and the MDCK cells were stained using 1% crystal violet (Sigma). The stained MDCK cells were then washed with distilled water to remove the crystal violet, virus plaques were counted and PFU per mL titer was calculated using the number of colonies and the dilution factor. The limit of detection for viral plaque titers was 50 PFU/mL.

**Neutralization assays.** A virus neutralization assay was used to identify the presence of influenza virus-specific neutralizing antibodies [23]. The protocol was adapted from the WHO influenza surveillance manual [24]. Briefly, equal amounts of sera from each mouse within a vaccination group were combined and heat-inactivated for 30 min at 56°C. MDCK cells were grown in a 96-well flat bottom plate until they had reached 95–100% confluency. Antibodies were diluted in ½-log increments with serum-free media and incubated with 100x $TCID_{50}$ for 1 h. The antibody-virus mixture was added to serum-free DMEM-washed MDCK cells in the 96-well plate. After 2 h, the MDCK cells were washed with serum-free DMEM. Next, 0.02 mL of DMEM with antibiotics/antimycotics (Gibco) and 2.0 ug/mL of TPCK trypsin (Thermo Fisher) was added to each of the 96 wells. The cell monolayers in the back-titration control wells were checked daily until cytopathic effect (CPE). After three days, 0.05 uL of media/well was removed and used in an HA assay to identify the presence of the virus. The remaining media in each well was removed, and the MDCK cells were fixed with 10% buffered formalin (Fisher Scientific) for 15 min. The formalin was then discarded, and the MDCK cells were

stained using 1% crystal violet (Sigma). The MDCK cells were then washed with distilled water to remove the crystal violet. CPE was defined as the lysis (dissolution) of the host cell monolayer.

## Statistical analysis

Statistical significance was defined as a $p < 0.05$. For HAI and neutralization assays, a titer of 1:5 was used as the limit of detection. The limit of detection for viral plaque titers was 50 pfu/mL. A one-way ANOVA with Tukey's ad-hoc test was used for the HAI, neutralization, and plaque assays. A two-way ANOVA with Tukey's ad-hoc test was used for weight loss with a statistical significance defined as a $p < 0.05$.

## Author Contributions

**Conceptualization:** Z. Beau Reneer, Harrison C. Bergeron, Ralph A. Tripp.

**Data curation:** Z. Beau Reneer, Harrison C. Bergeron, Stephen Reynolds.

**Formal analysis:** Z. Beau Reneer, Harrison C. Bergeron, Stephen Reynolds, Ralph A. Tripp.

**Funding acquisition:** Ralph A. Tripp.

**Investigation:** Ralph A. Tripp.

**Methodology:** Z. Beau Reneer, Elena Thornhill-Wadolowski, Lan Feng, Marcin Bugno, Agnieszka D. Truax, Ralph A. Tripp.

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
