## [Decision Letter · Decision Letter 0]

7 Dec 2023

PONE-D-23-35405mRNA Vaccines Encoding Influenza Virus Hemagglutinin (HA) Elicits Immunity in Mice from Influenza A Virus ChallengePLOS ONE

Dear Dr. Reneer,

Thank you for submitting your manuscript to PLOS ONE. After careful consideration, we feel that it has merit but does not fully meet PLOS ONE’s publication criteria as it currently stands. Therefore, we invite you to submit a revised version of the manuscript that addresses the points raised during the review process. During the revision process, please pay attention to the comments regarding overall experimental design (and impact of these findings), use of the challenge virus, and interpretation of findings in the overall context of the project.

We look forward to receiving your revised manuscript.

Kind regards,

Victor C Huber

Academic Editor

PLOS ONE

Journal Requirements:

3. To comply with PLOS ONE submissions requirements, in your Methods section, please provide additional information regarding the experiments involving animals and ensure you have included details on (1) methods of sacrifice,  and (2) efforts to alleviate suffering.

4. Thank you for stating the following financial disclosure: "Immorna-funded the research study."

5. Thank you for stating the following in the Competing Interests section: "The authors Thornhill-Wadolowski, Feng, Bugno, and Truax are employees of Immorna who provided the mRNA-LNP vaccines."   

We note that one or more of the authors are employed by a commercial company: Immorna 

6. We note that you have indicated that data from this study are available upon request. PLOS only allows data to be available upon request if there are legal or ethical restrictions on sharing data publicly. For more information on unacceptable data access restrictions, please see http://journals.plos.org/plosone/s/data-availability#loc-unacceptable-data-access-restrictions. 

7. PLOS requires an ORCID iD for the corresponding author in Editorial Manager on papers submitted after December 6th, 2016. Please ensure that you have an ORCID iD and that it is validated in Editorial Manager. To do this, go to ‘Update my Information’ (in the upper left-hand corner of the main menu), and click on the Fetch/Validate link next to the ORCID field. This will take you to the ORCID site and allow you to create a new iD or authenticate a pre-existing iD in Editorial Manager. Please see the following video for instructions on linking an ORCID iD to your Editorial Manager account: https://www.youtube.com/watch?v=_xcclfuvtxQ

8. We note that you have included the phrase “data not shown” in your manuscript. Unfortunately, this does not meet our data sharing requirements. PLOS does not permit references to inaccessible data. We require that authors provide all relevant data within the paper, Supporting Information files, or in an acceptable, public repository. Please add a citation to support this phrase or upload the data that corresponds with these findings to a stable repository (such as Figshare or Dryad) and provide and URLs, DOIs, or accession numbers that may be used to access these data. Or, if the data are not a core part of the research being presented in your study, we ask that you remove the phrase that refers to these data.

Reviewers' comments:

Reviewer's Responses to Questions

**Comments to the Author**

1. Is the manuscript technically sound, and do the data support the conclusions?

Reviewer #1: Yes

Reviewer #2: Yes

2. Has the statistical analysis been performed appropriately and rigorously? 

Reviewer #1: Yes

Reviewer #2: Yes

3. Have the authors made all data underlying the findings in their manuscript fully available?

Reviewer #1: Yes

Reviewer #2: Yes

4. Is the manuscript presented in an intelligible fashion and written in standard English?

Reviewer #1: Yes

Reviewer #2: Yes

5. Review Comments to the Author

Reviewer #1: Summary: The authors immunised mice with a quadrivalent influenza mRNA vaccine or individual components. They measured antibody responses against a broad range of viruses and protection against H1N1 challenge.

Major comments:

1. The introduction needs expanding on other studies that have used mRNA vaccines for influenza to put the work into context. At present it is one paragraph of introduction and then another paragraph summarising the current studies.

2. The work could be more ambitious in scope – for example challenge with a different virus to show the benefits of combining the mRNA vaccines. The cross neutralisation data is of interest, but the rest of the manuscript is light on experiments – it could be compressed to 2 or maybe 3 figures at the moment.

Minor comments:

1. Rather than saying group 1-10 in numbers put what they received on the graphs, it is very hard to read.

2. Why are only some groups challenged with virus

3. Y axis scale is wrong on the weight loss graph.

4. Figures 3 and 4 put the virus strains above each graph

5. Figure 5 present as individual points

Reviewer #2: This manuscript reports results of investigation of the protection of single and quadrivalent mRNA vaccination. Four different influenza viruses including two influenza A viruses (IAV) and two influenza B (IBV) were designed to protect mice from influenza virus strain attack. Then, antibody tites, viral lung titers, number of lymphocytes were detected.

This is a interesting research. It provide data support for future applications.

There are some scientific questions that need to be clarified in the results or discussion:

1.Materials and methods, How many animals were used in this experiment? How many animals were used in each group and do they all have pathogenic symptoms?

2.Why did the author use A/California/04/2009 as the challenge strain? Is this strain still representative at present? What would happen if the current epidemic strain were used for the challenge?

3.Table 2, A/Dar/21 and B/Phu/13 strains produced low antibody titers, especially group 8, Why?

4.Figure 1, Why the weight value of group 3 was low, even lower than the control (group 1)?

5.Figure 2, why the vaccine provided complete protection in groups 2 and 6? Is there any correlation between immune and challenge strains?

6.Figure 3 D and Figure 4 D, it was low levels of HAI antibodies immunized with the Phuket/3073/2013 strain in group 5, why?

7.Figure 4 D and H, The mixed vaccine also did not produce significant antibodies. Why? It is necessary to discuss the characteristics of the strain.

8.Figure 5, where are 4 and 5 groups?

9.Figure 5, I think that it may not provide significant result. The number of T cells would significant change only in consumptive or critical diseases. It is more meaningful to analyze the number of T cells that can secrete specific cytokines.

6. PLOS authors have the option to publish the peer review history of their article (<a href="https://journals.plos.org/plosone/s/editorial-and-peer-review-process#loc-peer-revie

---

## [Author Response · Author response to Decision Letter 0]

8 Dec 2023

Reviewer 1:

Major comments:

1. The introduction needs expanding on other studies that have used mRNA vaccines for influenza to put the work into context. At present it is one paragraph of introduction and then another paragraph summarising the current studies.

We thank the reviewer for their comment. We have added a paragraph discussing other mRNA influenza vaccine studies.

2. The work could be more ambitious in scope – for example challenge with a different virus to show the benefits of combining the mRNA vaccines. The cross neutralization data is of interest, but the rest of the manuscript is light on experiments – it could be compressed to 2 or maybe 3 figures at the moment.

We thank the reviewer for their comment. We plan to expand our experiments with future studies, but we feel that the number of figures and tables is appropriate for this manuscript. We will take this comment into consideration to improve the studies for our future manuscripts. 

Minor comments:

1. Rather than saying group 1-10 in numbers put what they received on the graphs, it is very hard to read.

We thank the reviewer for their comment. We have added labels for clarification.

2. Why are only some groups challenged with virus

We thank the reviewer for their comment. We had excluded Groups 4 and 5 from several figures due to mice in each group having to be sacrificed. We have included the weight loss data for the mice in for these groups in figure 1 and added clarification in lines 113-15 explaining these group exclusions from figure 2. 

3. Y axis scale is wrong on the weight loss graph.

We thank the reviewer for their comment. We have corrected this error.

4. Figures 3 and 4 put the virus strains above each graph

We thank the reviewer for their comment. We have added the stains for clarification.

5. Figure 5 present as individual points

We thank the reviewer for their comment. Due to several comments about this figure, we have completely removed the figure from the manuscript.

Reviewer 2:

1.Materials and methods, How many animals were used in this experiment? How many animals were used in each group and do they all have pathogenic symptoms?

We thank the reviewer for their comment. We have added information on lines 289-299.

2.Why did the author use A/California/04/2009 as the challenge strain? Is this strain still representative at present? What would happen if the current epidemic strain were used for the challenge?

A/California/04/2009 is a well-established challenge virus for influenza vaccine studies (doi: 10.1016/j.antiviral.2011.02.001, doi: 10.1128/JVI.00076-13). A/California/04/2009 is a pH1N1 virus that is the same lineage that still circulates today. In our study, we wanted to challenge with an antigenically distinct H1N1 virus to ensure that our vaccine was electing a broadly cross-reactive antibody response and not a narrow antibody response only to the HA antigen that was used in the vaccine. 

3.Table 2, A/Dar/21 and B/Phu/13 strains produced low antibody titers, especially group 8, Why?

We thank the reviewer for their comment. H3N2 and Influenza B strains are much less immunogenic than H1N1 strains. Group 8 is a single vaccination while all other groups received 2 vaccinations which is likely the reason for the lower titers.

4.Figure 1, Why the weight value of group 3 was low, even lower than the control (group 1)?

We thank the reviewer for their comment. While it appears that there is slight difference on days 4 and 5 between these groups (as well as groups 4 and 5 that were added to this figure), these differences are not statistically significant. 

5.Figure 2, why the vaccine provided complete protection in groups 2 and 6? Is there any correlation between immune and challenge strains?

We thank the reviewer for their comment. Group 2 and 6 had high HAI titers to Cal/09 (Figure 4). HAI titers are often highly correlated with low lung viral titers. Groups 2 and 6 likely had Ab titers high enough to limit viral replication to below the limit of detection for the plaque assay. 

6.Figure 3 D and Figure 4 D, it was low levels of HAI antibodies immunized with the Phuket/3073/2013 strain in group 5, why?

We thank the reviewer for their comment. There is a single amino acid mismatch between the vaccination strain and the Phuket/3073/2013 virus used in figures 3 and 4. This is stated in lines 243-245.

7.Figure 4 D and H, The mixed vaccine also did not produce significant antibodies. Why? It is necessary to discuss the characteristics of the strain.

We thank the reviewer for their comment. The amino acid mutation is likely causing lower titers in panel D. The results in panel H are showing an antigenically distinct B/Yamagata strain. These results are similar to the antigenically distinct B/Victoria strain in panel G. None of the mixed vaccine produced high antibody titers to any of the antigenically distinct IBVs. This is likely due to the low immunogenicity of IBV HA proteins. 

8.Figure 5, where are 4 and 5 groups?

We thank the reviewer for their comment. Due to several comments about this figure, we have completely removed the figure from the manuscript.

9.Figure 5, I think that it may not provide significant result. The number of T cells would significant change only in consumptive or critical diseases. It is more meaningful to analyze the number of T cells that can secrete specific cytokines.

We thank the reviewer for their comment. Due to several comments about this figure, we have completely removed the figure from the manuscript.

---

## [Decision Letter · Decision Letter 1]

14 Jan 2024

mRNA Vaccines Encoding Influenza Virus Hemagglutinin (HA) Elicits Immunity in Mice from Influenza A Virus Challenge

PONE-D-23-35405R1

Dear Dr. Reneer,

Thank you for responding to the comments from the previous review and for presenting data that contributes to the research on mRNA vaccines against influenza viruses.  No further modifications are needed as there were no additional comments regarding improvements for the data presented.

We’re pleased to inform you that your manuscript has been judged scientifically suitable for publication and will be formally accepted for publication once it meets all outstanding technical requirements.  

Kind regards,

Victor C Huber

Academic Editor

PLOS ONE

Additional Editor Comments (optional):

Reviewers' comments:

Reviewer's Responses to Questions

**Comments to the Author**

1. If the authors have adequately addressed your comments raised in a previous round of review and you feel that this manuscript is now acceptable for publication, you may indicate that here to bypass the “Comments to the Author” section, enter your conflict of interest statement in the “Confidential to Editor” section, and submit your "Accept" recommendation.

Reviewer #1: (No Response)

Reviewer #2: All comments have been addressed

2. Is the manuscript technically sound, and do the data support the conclusions?

Reviewer #1: No

Reviewer #2: Yes

3. Has the statistical analysis been performed appropriately and rigorously? 

Reviewer #1: No

Reviewer #2: Yes

4. Have the authors made all data underlying the findings in their manuscript fully available?

Reviewer #1: No

Reviewer #2: Yes

5. Is the manuscript presented in an intelligible fashion and written in standard English?

Reviewer #1: No

Reviewer #2: Yes

6. Review Comments to the Author

Reviewer #1: My main concern was there was not enough substance to the paper. The authors have added no new experiments.

Reviewer #2: (No Response)

7. PLOS authors have the option to publish the peer review history of their article (what does this mean?). If published, this will include your full peer review and any attached files.

Reviewer #1: No

Reviewer #2: No

---

## [Editor Report · Acceptance letter]

8 Apr 2024

PONE-D-23-35405R1 

PLOS ONE

Dear Dr. Reneer, 

I'm pleased to inform you that your manuscript has been deemed suitable for publication in PLOS ONE. Congratulations! Your manuscript is now being handed over to our production team.

Kind regards, 

on behalf of

Dr. Victor C Huber 

Academic Editor

PLOS ONE